# Research collaboration and topic trends in Computer Science based on top active authors

Yan Wu[1], Srinivasan Venkatramanan[2] and Dah Ming Chiu[1]

[1] Department of Information Engineering, The Chinese University of Hong Kong, Hong Kong
[2] Virginia Bioinformatics Institute, Virginia Polytechnic Institute and State University (Virginia Tech), United States

## ABSTRACT

Academic publication metadata can be used to analyze the collaboration, productivity and hot topic trends of a research community. In this paper, we study a specific group of authors, namely the top active authors. They are defined as the top 1% authors with uninterrupted and continuous presence in scientific publications over a time window. We take the top active authors in the Computer Science (CS) community over different time windows in the past 50 years, and use them to analyze collaboration, productivity and topic trends. We show that (a) the top active authors are representative of the overall population; (b) the community is increasingly moving in the direction of Team Research, with increased level and degree of collaboration; and (c) the research topics are increasingly inter-related. By focusing on the top active authors, it helps visualize these trends better. Besides, the observations from top active authors also shed light on design of better evaluation framework and resource management for policy makers in academia.

Corresponding author
Dah Ming Chiu,
dmchiu@ie.cuhk.edu.hk

## INTRODUCTION

As a research field established in the 1960s (*Brookshear, 2011*), Computer Science has gone through rapid development and become a mature field. Much can be learned about the developments and trends in Computer Science by analyzing the publication metadata. In this study, we take a particular approach, by focusing on analyzing the Top Active Authors in the field. We define top active authors to be the 1% authors with uninterrupted and continuous presence in scientific publications over a time window. The definition of Top Active Authors is based on the term "UCP author", which was defined by *Ioannidis, Boyack & Klavans (2014)* in their study of publication metadata obtained from Scopus in a specific time window of 16 years. During the period from 1996 to 2011, they noted that the number of authors who published papers every year without gap amounts to about 1% of all authors; and these UCP authors coauthored a much larger percentage of papers and amassed a high percentage of the total citations, compared to the average researcher. Therefore, we might treat top active authors as the core of the community for the given time window. By analyzing their activities, we may get insights into the major trends of the whole community.

In our study, we further explore the nature and extent of collaborative efforts by the top active authors in comparison with average authors. Recently, it was pointed out by *Wuchty, Jones & Uzzi (2007)* that "Team Science" is an important trend in how research is carried out. The phenomenon is manifested in the steady increase in the number of coauthors per paper. Since this trend exists not only in science but also in other research fields, we can refer to it as "Team Research". The "team" in Team Research may correspond to an organized group within an organization (a research lab), or collaborative partnership between researchers in different organizations and countries. From the collaboration patterns of top active authors, we can get more insights about Team Research, in particular its relation to research productivity. Besides, by observing the research topics that top active authors are working on, we can also get a sense of the general research topic trends in the whole community.

Our goal goes beyond simple data analysis in this work. We hope the observations from top active authors can not only show the general trends in doing research in the academic community, but also provide insights for policy researchers and policy makers in academia. For example, the comparison between top active authors and average authors might shed light on the shortcomings of current author evaluation framework, and help develop different evaluation metrics for different author types. And the general trends in research collaboration and topic may also help policy makers adjust resource allocations at different periods and do better resource management.

## RELATED WORK

Research collaboration has been long studied in the scientometrics field. For example, *Katz (1994)* examined the geographical effects on intra-national scientific collaboration and demonstrated that research collaboration decreases exponentially with the distance separating the collaborative partners. *Wagner & Leydesdorff (2005)* showed that international collaboration is a self-organizing network, and its growth can be explained based on the organizing principle of preferential attachment. There are also a lot of works focusing on the statistically significant increase in the amount of research collaboration in this field. For example, *O'Brien (2012)* showed the overall increase in the number of coauthored articles in the literature; *Wuchty, Jones & Uzzi (2007)* coined the term "team science" and showed the increasing dominance of team science in the production of knowledge; and *Uddin, Hossain & Rasmussen (2013)* studied the network effects on authors' collaboration behaviors.

The authors' research collaboration behavior also attracted attentions from researchers studying networks, and most of the studies were based on the classic work from *Newman (2001)*. He treated the coauthor network as a special example of a social network and studied the structural properties of such a network. Researchers have also studied the structural evolution of a collaboration network. For instance, *Kunegis, Fay & Bauckhage (2010)* analyzed the eigenvector evolution of the coauthor network and proposed a spectral evolution model to show the change of coauthor structures; *Huang et al. (2008)* proposed a stochastic Poisson model with optimization tree, which can efficiently predict

the increment of collaboration based on local neighborhood structure; *Pan & Saramäki (2012)* and *Ke & Ahn (2014)* studied the relationship between tie strength and network topology in coauthor network and found its difference with social networks in general.

Besides, research collaboration has always been an important component when considering policies towards research. Many works focusing on research policy studies have tried to evaluate the efficiency of resource management in system based on researchers' collaboration behavior. Classic works in this field include *Katz & Martin (1997)*, where the authors studied research collaboration at different levels and argued for a more symmetrical approach in comparing the costs of collaboration with the benefits; and *Lee & Bozeman (2005)*, where the authors investigated the impact of research collaboration on scientific productivity and showed the different impacts at individual level and community level. They also proposed considering collaborations in terms of the extent to which resources fit research needs. There are also works based on surveys of individual researchers' opinions on research collaboration such as *Melin (2000)*, where the author suggested that research policy should provide financial and organizational possibilities for the researchers to establish joint ventures and also fund projects on a team or network basis.

Another group of papers related to our work studied the factors that may influence productivity or authors' research behavior. For example, *Gingras et al. (2008)* showed the effects of aging on researchers' publication patterns and described researchers' publication style during different stages of career. *Petersen et al. (2012)* found the existence of the Matthew effect in academic publishing, which may favor senior and experienced researchers. Finally, *Ioannidis, Boyack & Klavans (2014)* was the first to introduce the notion of uninterrupted and continuous presence as a way to identify a set of core authors in a research community, and showed the dominance of these authors in the production of academic outputs.

Most of the previous works (except *Ioannidis, Boyack & Klavans (2014)*) analyze the entire population of a community. By focusing on top active authors, which is a much smaller, but important and representative subset of the overall population, we are able to find more results about trends in research collaboration (team research), its relationship to research productivity, and the evolution of research topics and focus as well.

## MATERIALS & METHODS

Our data is collected from Microsoft Academic Search (MAS) (http://academic.research.microsoft.com/). MAS gathers bibliographic information from the principal scientific publishing services covering papers from 1700 to present, and uses its own classification scheme based on 15 research fields and more than 200 subdomains to classify different papers (*Orduña-Malea et al., 2014*). For example, the 15 fields include Computer Science, Physics, and Mathematics, and papers in the Computer Science field can be further categorized into 24 subdomains such as "Databases", "Machine learning and Pattern Recognition", "Networks and Communications" and so on.[1] Each paper is labelled with a unique numerical ID; its metadata includes paper title, author list, publication year,

[1] Some papers are only classified as Computer Science papers, but not categorized into any subdomain.

**Table 1** **Dataset description.**

| | |
|---|---|
| Field coverage | Computer Science |
| Time coverage (year) | 1960–2009 |
| #papers | 2,698,044 |
| #authors | 1,393,143 |
| #paper-author mapping links | 6,643,575 |

publication venue and a reference list. Likewise, authors are maintained as another type of object. Each author is labelled with a unique numerical ID as well; its metadata includes current affiliation and publication history. An author's research field and research subdomains in that field can be obtained from his publication history. We choose the Computer Science field, which seems most complete (and we are most familiar with) for a case study in this paper. The same methodology can be applied to data from other sources, pertaining to other scientific disciplines.

Based on the mapping information of papers and authors, we can obtain the detailed evolution trend of author collaboration and author productivity for both top active authors and average authors. This can show us the general trend in the community as a whole, and also the difference between top active authors and average authors. Besides, since our metadata comes with classification of each paper into a research subdomain in Computer Science, it is possible to tell the subdomains each top active author works in. Given the moderate size for the set of all top active authors, it is possible to apply graph clustering algorithms to find the collaboration clusters for top active authors in Computer Science over time, and characterize these clusters in terms of the major subdomains they are working on. Such analysis will show us the research topic trend in Computer Science.

Considering the fact that the data for the earlier and the most recent years are less complete, we take the data in the 50-year window [1960, 2009] for our analysis in this paper.[2] Note that although *Orduña-Malea et al. (2014)* pointed out the limitation of the data from MAS, which includes the decreasing update rate and incomplete indexation of all papers recently, the limitation and misleading information lies mostly in the data starting from 2010, as the authors indicated. Therefore, we believe the data in the time window we focus on is relatively reliable, and will not affect the validity of our results. We also filter out paper records without publication year or author information for this study. Table 1 presents a general description of our dataset. The more detailed evolution of the dataset is shown in Fig. 1, where we plot the annual number of active authors and papers each year in the time window [1960, 2009]. The rapid expansion of the Computer Science field can be observed clearly.

[2] The data from MAS was lastly collected on July 31, 2012.

# RESULTS

## Comparing top active authors and average authors

In this section, we analyze and compare the collaboration levels and patterns of top active authors versus average authors, as well as their productivity. We take one active window as

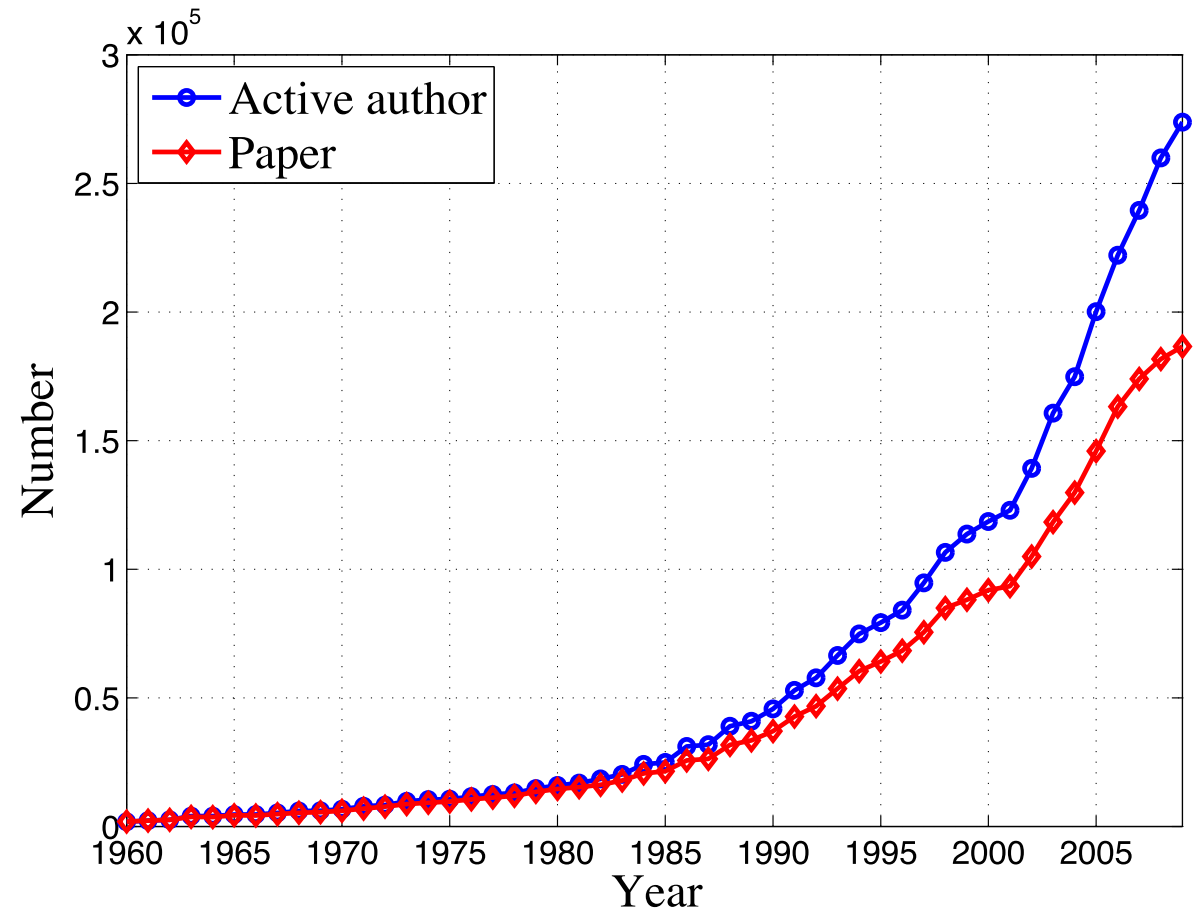

**Figure 1** **Number of active authors and papers each year in the time window [1960, 2009].**

an example, and show the detailed comparison across different years in that window. Since results in other active windows are similar, we then show brief results across different active windows, instead of the detailed comparison in each active window.

### *Top active authors*

*Ioannidis, Boyack & Klavans (2014)* defined "UCP authors" by considering a specific window of years, from 1996 to 2011, and observed that there are about 1% such authors. For our purposes, we define the top active authors based on the UCP metric. For each year to be used as the start of a window, we find the top 1% authors in terms of the length of uninterrupted and continuous presence from that starting year. This gives rise to an active window size for top active authors for each year. For example, starting from year 1988, the active window size needs to be set as 8 (which means the ending year of that active window is 1995), in order to make the percentage of top active authors among authors with at least one publication in that window around 1%. Smaller window size will lead to a higher percentage than 1% while larger window size will lead to a lower percentage. For clarity, we show an example of top active author versus non-top active author in the active window [1988, 1995] in Fig. 2, where their annual number of publications during that period is plotted.

With this definition, it is observed that the active window size required to be counted as a top active author is different for each starting year. In fact, this active window size is growing steadily over the years, as shown in Fig. 3. This certainly correlates well with our impression that the top authors are becoming more and more active. The number of years required to become a top active author starting from 1996 is around 11, which is a little less than that found in *Ioannidis, Boyack & Klavans (2014)*. This is not too surprising considering the research field and dataset studied are both different. But the result is in the same ball park.

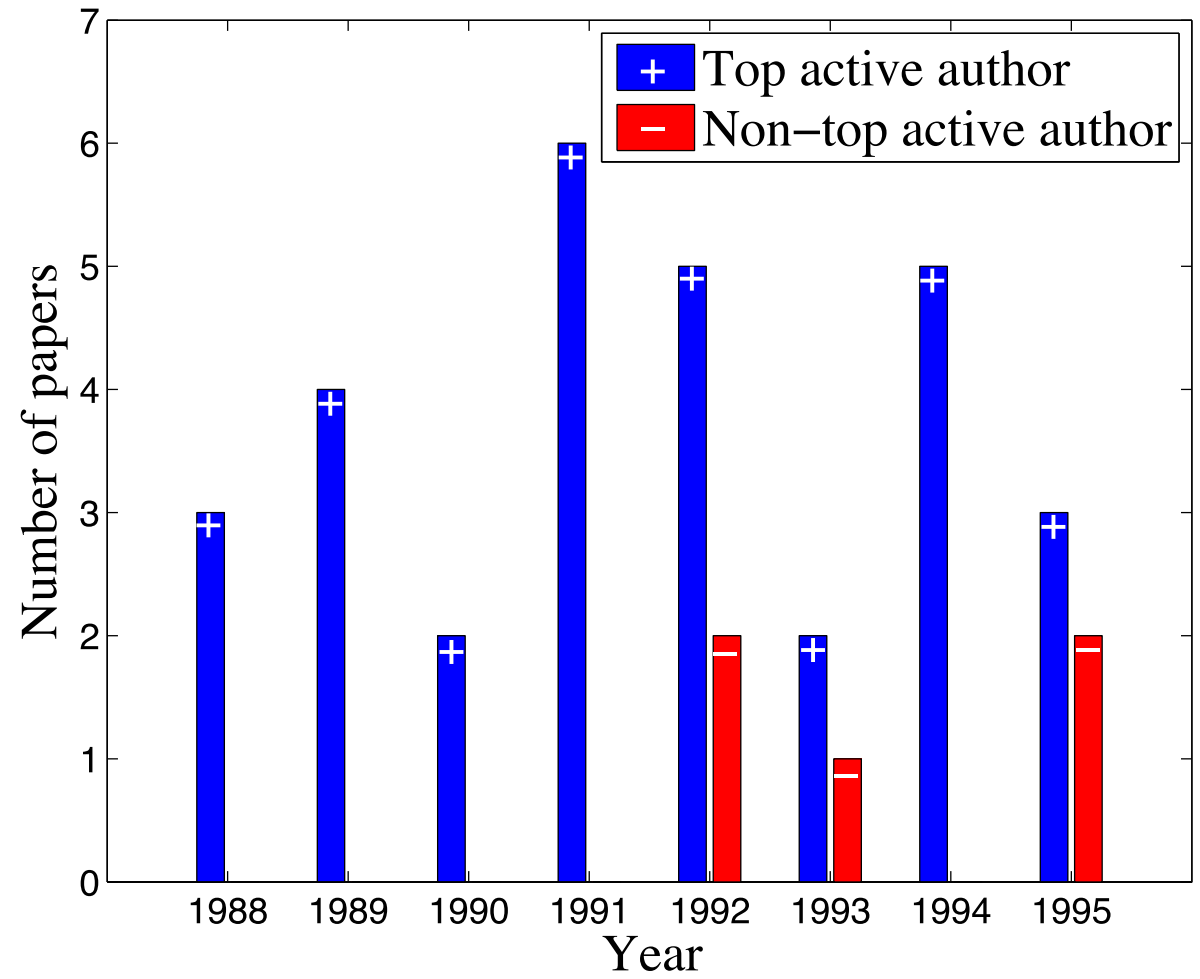

**Figure 2** Example of top active author versus non-top active author in the active window [1988, 1995].

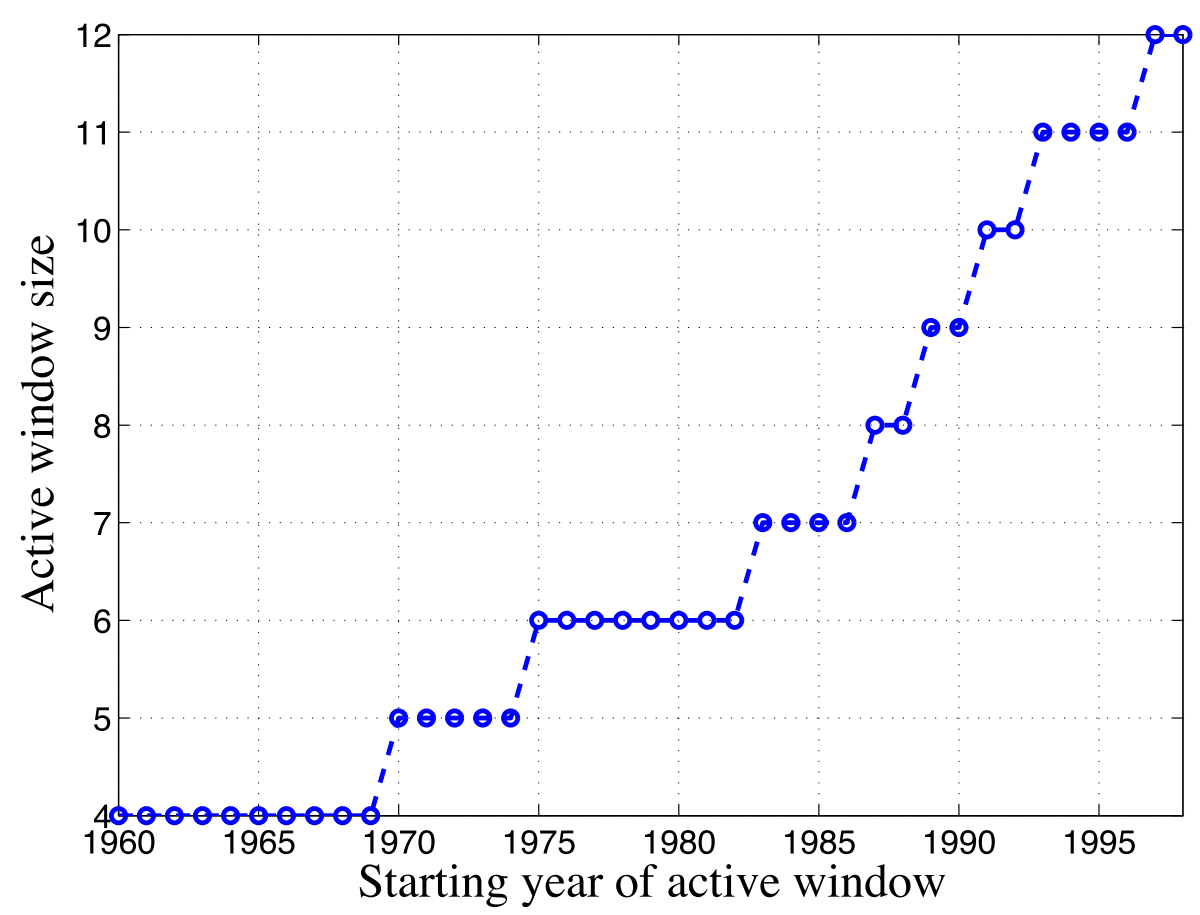

**Figure 3** Change of active window size.

### *Comparison on collaboration*

We first compare the collaboration patterns of top active authors with that of average authors. Top active authors is the author set including all top active authors in an active window, while "average authors" is the author set including any author with at least one publication in an active window. So average authors is a superset of top active authors. We take the active year window [1988, 1995] for comparison. We compare the nature and extent of collaboration, such as the size of coauthors set, collaboration strength and team connectivity, and then the productivity. We will show the brief comparison results across different active windows later.

One important measure for the extent of collaboration of an author, obviously, is the size of the coauthors set. Figure 4 shows the average number of coauthors per author for top active authors and average authors on an annual basis. Here the coauthors include top active coauthors and non-top active coauthors. The figure shows that top active authors

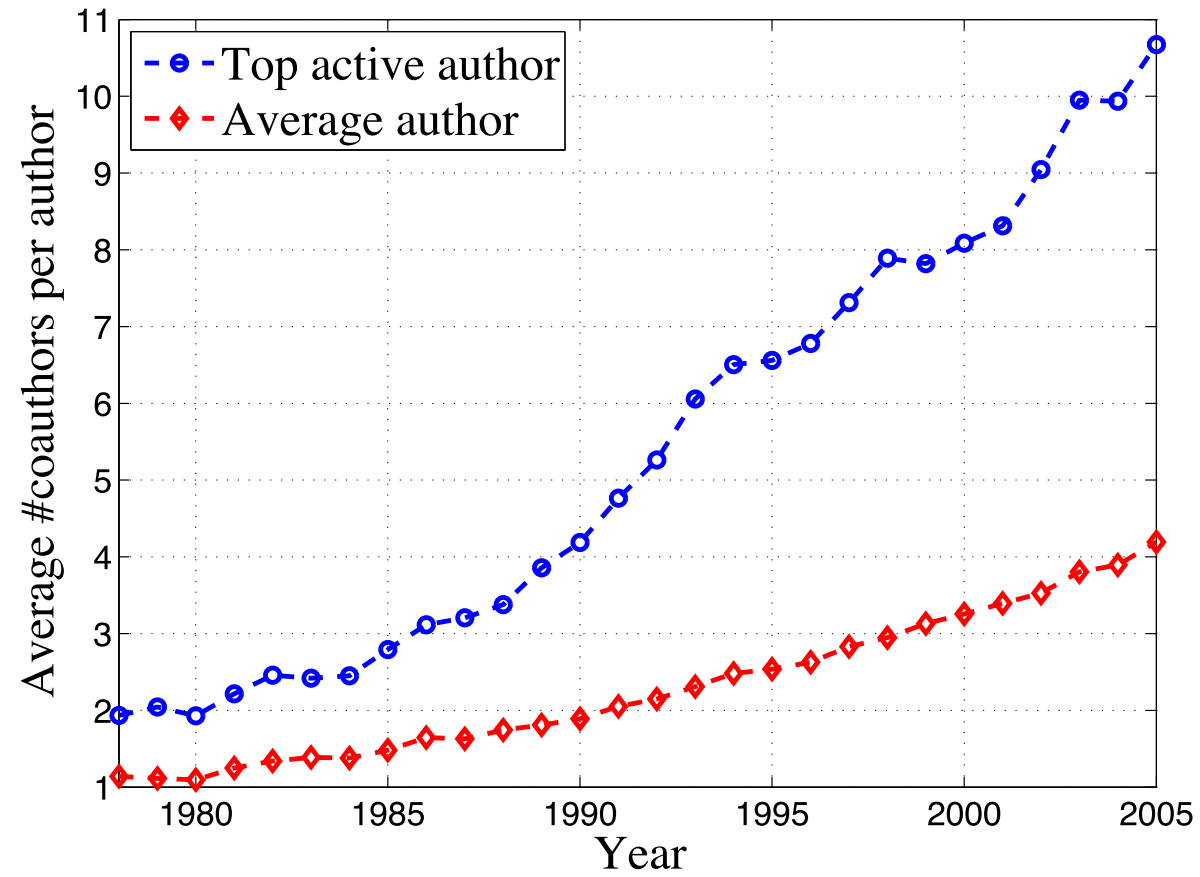

**Figure 4** **Comparison of coauthors set size between top active authors and average authors.**

generally have more coauthors than average authors. Moreover, the top active authors also have a significantly higher increase rate of coauthors, although more collaboration is the trend for the whole community (*O'Brien, 2012*; *Wuchty, Jones & Uzzi, 2007*).

A likely reason for the much higher number of coauthors for top active authors is directly due to "team science" and the growth in team size. As the collaboration pattern in a team is often hierarchical, and the top active authors are more likely at the root of the hierarchy, they would naturally have more collaborators and benefit from growth of team sizes. If we assumed the coauthor network is built by preferential attachment (*Lee et al., 2010*), we would reach the same conclusion for top active authors. To further understand the collaboration pattern by top active authors, we also show the coauthors set size of the same set of top active authors during their pre top-active period and post top-active period for ten years in Fig. 4. It is clear that even before and after their top-active period, top active authors tend to have more coauthors. The difference between top active authors and average authors in the ten years before their top-active period is not so much as that in later periods. But there still exists slight advantage to top active authors. This indicates that in order to become top active authors, it is important for authors to build and expand their research teams in the very beginning. The further growth of the number of coauthors in the post top-active window is likely due to the reputation and connections they accumulated during their top-active period.

In a social network, while the number of friends may show the size of one's social connections, the length of the friendship one keeps with others can better reflect the extent of one's influence in his social network. Similarly, in the study of research collaboration, we can also use the collaboration length between one and his coauthors to represent one's influence in his research community. Again we compare top active authors with average authors using the active window [1988, 1995]. Figure 5A shows the distribution of collaboration length in the 8-year active window for top active authors and average authors. We observe that for both top active authors and average authors, more than half of the collaboration links exist for only one year, which shows the dominance of

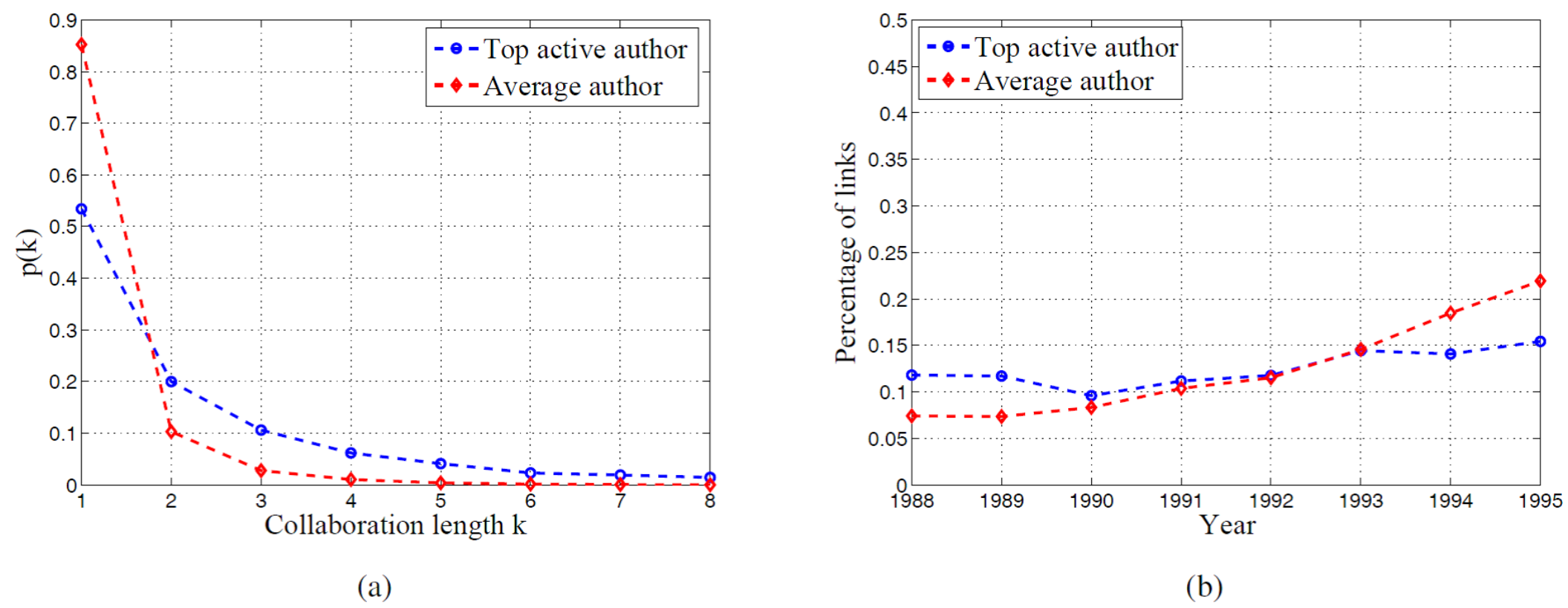

**Figure 5 Comparison of collaboration length between top active authors and average authors.** (A) is the distribution of collaboration length. (B) is the distribution of one-year link.

**Table 2 Analysis on connected components.**

| Statistical measurement | Top active author | Average author |
|---|---|---|
| #nodes in network | 2,317 | 212,527 |
| #links in network | 5,677 | 410,606 |
| #connected components | 61 | 25,201 |
| #nodes in giant component | 2,170 | 135,391 |

short-term research teams. However, top active authors are more likely to have longer collaboration relationship with others. This indicates that although top active authors have a rapid expansion rate of coauthors, there still exist some stable collaboration links. For the transient links which last for only one year, the distribution of the year in which the one-year collaboration happens is plotted in Fig. 5B. While it is almost uniformly distributed in the 8 years for top active authors, it is left skewed for average authors. Top active authors keep a regular proportion of transient collaboration links, while the average authors have more short-term collaboration links in recent years, which may be the result of rapid increase of paper publishing over years.

Next, we analyze the structure of the coauthor networks built in the 8-year window by top active authors and average authors respectively. Here the coauthor network of top active authors contains the collaboration between top active authors only and the average author coauthor network consists of all the authors with publications in the specific time window. For simplicity, we have removed authors with no coauthors (single nodes only) in the two networks. The result is shown in Table 2, where we focus on the analysis of connected components in the two networks. The number of connected components is a lower bound to the estimated number of clusters in the coauthor network. It reflects the connectivity in the network as a whole. As shown by Table 2, top active authors are more connected with each other while for average authors, small teams are more popular.

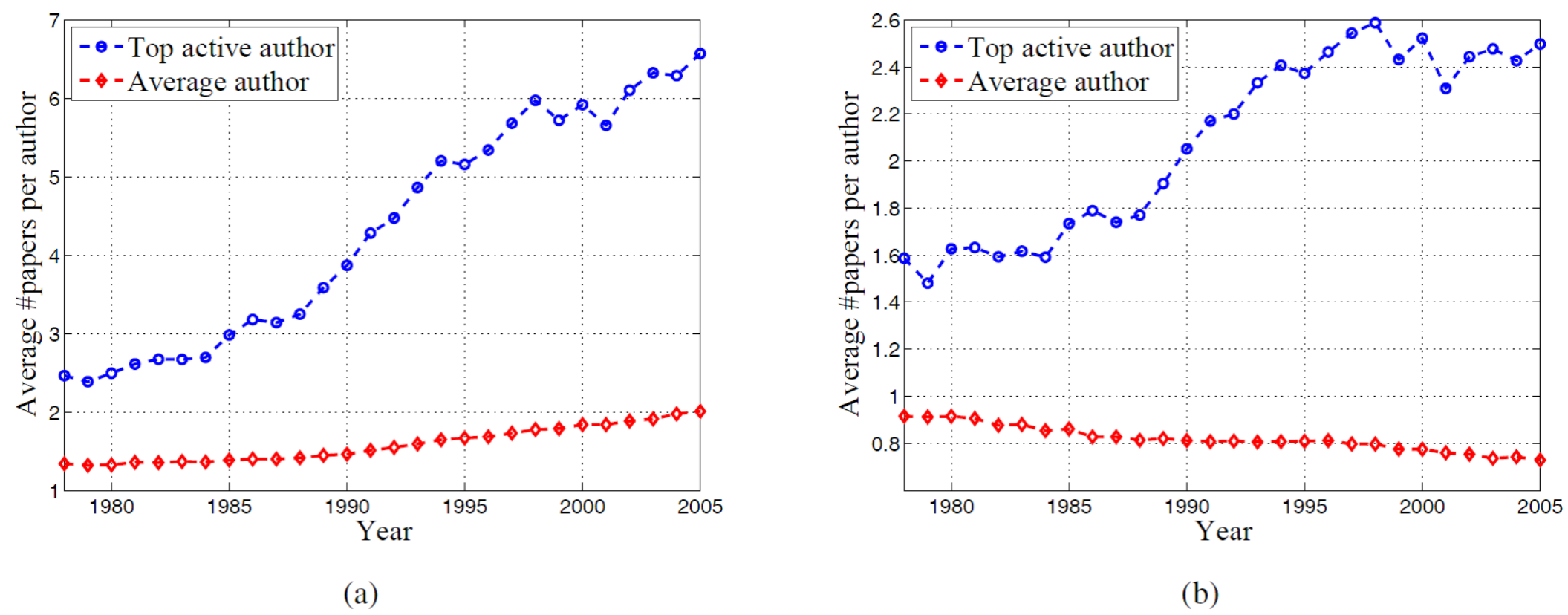

**Figure 6 Comparison of productivity for top active authors and average authors.** (A) is the comparison of IP. (B) is the comparison of CP.

### *Comparison on productivity*

Besides the collaboration patterns, a more direct assessment of an author's activity in the community is productivity, which is often reflected by the annual publication rate of an author. Before going to the detailed discussion of productivity, we define two notations first: individual productivity (IP) and community productivity (CP). IP is the annual number of claimed papers per author. Thus IP is incremented for an author every time his name appears in a paper. CP, on the other hand, is based on the fractional contribution of each coauthor towards a paper (equal division assumed). CP counts each paper only once, while IP counts each paper $n$ times when there are $n$ coauthors. Figure 6 shows the comparison of IP and CP for top active authors and average authors.

Similar to the comparison of coauthors, we also include productivity behaviors in the ten years before and after the top-active period. For the comparison of IP, we can see that IP almost doubles in the 28 years for average authors, while it is much more than doubled for top active authors. This can be partially explained by the different coauthors set sizes of top active authors and average authors. Different from CP, the contribution of coauthors can help increase one's IP. We can see in Fig. 4 that while the annual number of coauthors for average authors increases from 1 to 4 in the 28 years, it increases from 2 to 11 for top active authors in the same period. Such a rapid expansion of collaboration thus inevitably leads to more productivity for top active authors. For CP, there is a slight decreasing trend for average authors, whereas for top active authors, the trend is increasing over the 28 years. This shows that although the top active authors are consistently increasing their productivity, whether measured by IP or CP, the productivity (CP) of the average authors are actually decreasing. This phenomenon was also observed and discussed in the context of team science (*Wuchty, Jones & Uzzi, 2007*).

### *Comparison across different active windows*

Now we show the brief comparison results between top active authors and average authors across different active windows. As previously, we do the comparison on collaboration and productivity. For collaboration, the average annual number of coauthors per author

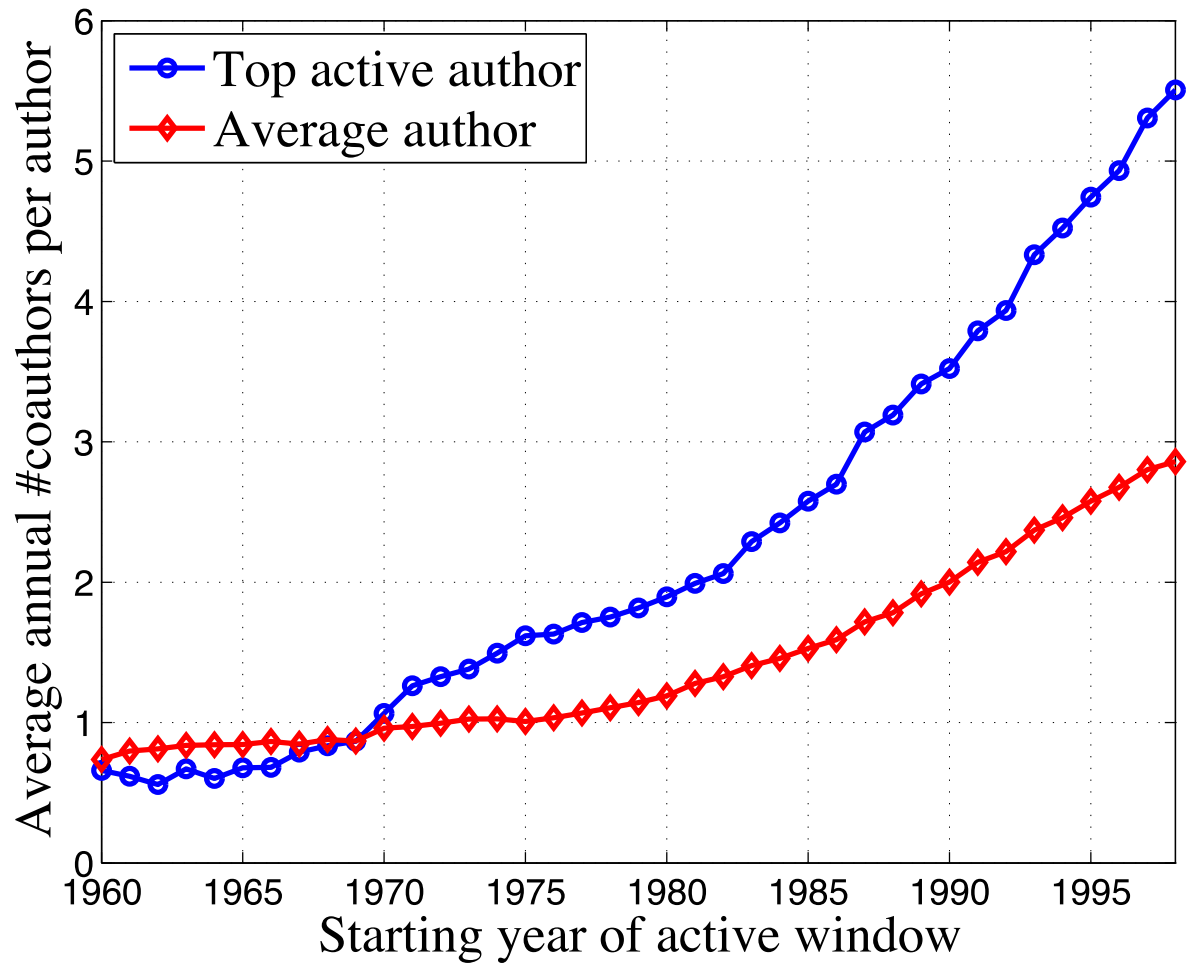

**Figure 7 Comparison of coauthors set size between top active authors and average authors across different active windows.**

in each active window starting from different years is plotted in Fig. 7. The calculation is as follows: for each targeted author, we count his number of active years and number of distinct coauthors in one active window (That means if a coauthor collaborates multiple times with the targeted author in an active window, we only count him once.). With the two numbers, we obtain his annual number of coauthors in the active window. Then we do average on the targeted author set, and get the comparison result between top active authors and average authors. From Fig. 7 we see that in the earlier windows, there is not much difference between top active authors and average authors, they have coauthors with almost the same scale. However later, top active authors show great advantage over average authors in attracting coauthors. We see that in the last active window starting from 1998, the average annual number of coauthors per author is almost doubled for top active authors than average authors. For productivity, we also do the comparison on IP and CP. The comparison result is shown in Fig. 8, where we plot the average annual number of papers per author in each active window starting from different years. The calculation is similar to the previous calculation for coauthors. We see that for IP, while average authors behave consistently over different active windows, top active authors show a rapidly increasing trend. For CP, while average authors show a decreasing tendency, top active authors are still able to achieve increasing productivity, although the increase rate is not that large when compared to the IP case. In summary, the comparison results between top active authors and average authors across different active windows are consistent with our previous findings in one active window.

The analysis and comparison of top active authors and average authors, on both collaboration and productivity, give us further insights into the trend of team research. Top active authors are able to achieve more sustained research activity, much higher level of research output, and accelerated growth in research output. This may be partially explained by their ability to build research teams, as well as to form extensive research

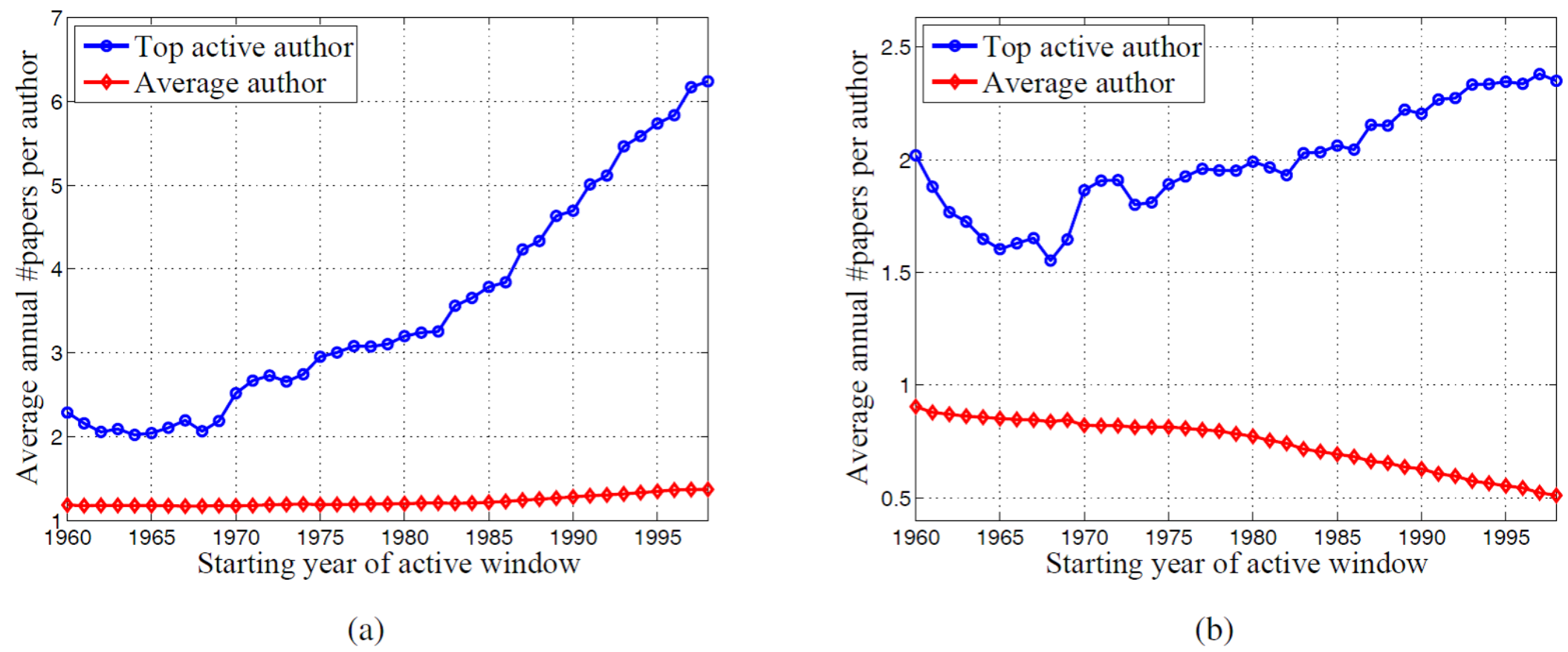

**Figure 8 Comparison of productivity for top active authors and average authors across different active windows.** (A) is the comparison of IP. (B) is the comparison of CP.

collaboration with a broader range of other authors including top active authors. In this regard, the top active authors might be treated as the core of the research community.

## Collaboration among top active authors

From our analysis above, top active authors can be considered as the core of their academic community. Their collaboration and communication plays a significant role in knowledge diffusion and research development in academia. And the evolution of their collaboration patterns also reflect the research trend in the whole community. Therefore, we take an analysis on the collaborations among those top active authors in this section.

For the study of top active author collaboration, two important aspects of the collaboration network is its topology and tie strength. While network topology demonstrates the way top active authors link with each other, tie strength indicates the closeness of the linked top active authors. Our focus for this study is thus the relationship between the tie strength and network topology of the coauthor network of top active authors. Previous studies on this have shown different results. *Granovetter (1973)* and *Onnela et al. (2007)* have already shown that in ordinary social networks, members usually communicate more frequently with other ones in the same community. Therefore in the communication network built by community members, if we define the tie strength in the network by their communication frequency, we can find that stronger ties exist mostly inside a community, while ties between different communities are usually weaker. However, in coauthor networks, it is shown in *Pan & Saramäki (2012)* and *Ke & Ahn (2014)* that on the contrary, the tie strength is usually much weaker among authors in the same community than authors in different communities. Then as the core of the academic community, for top active authors, what will be the case in their coauthor network?

Before we take a detailed analysis on this, we first give formal definitions on the measurement of tie strength between coauthors and network topology. Following the practice of *Newman (2001)*, we define tie strength, i.e., collaboration link weight between

author $i$ and one of his coauthor $j$ by

$$w_{ij} = \sum_{p} \frac{1}{n_p - 1}, \tag{1}$$

where $p$ is the set of papers that author $i$ and $j$ have coauthored, and $n_p$ is the number of authors of paper $p$. Therefore, for the coauthor network of top active authors, $w_{ij}$ is calculated based on all the coauthored papers by two top active authors in an active window. Note that the definition of the collaboration link weight here considers not only the collaboration length between two top active coauthors, but also their collaboration frequency.

For the network topology, we focus on the measurement of the similarity of two coauthors' neighborhood, which may reflect whether two coauthors are in the same community or not. So we define the neighborhood similarity of author $i$ and author $j$ by

$$O_{ij} = \frac{n_{ij}}{d_i - 1 + d_j - 1 - n_{ij}}, \tag{2}$$

where $d_i$ is the node degree of author $i$, $d_j$ is the node degree of author $j$, and $n_{ij}$ is the number of common neighbors of author $i$ and author $j$ in the coauthor network (*Onnela et al., 2007*) of top active authors. Thus $O_{ij}$ reflects the link overlap of two top active coauthors.

After the introduction of the above definitions, we then do analysis on the coauthor network of top active authors. We take two sets of top active authors in the active windows [1988, 1995] and [1998, 2009] respectively, and study the evolution by comparing the difference between them. As with the previous analysis, we build the two coauthor networks based on the existence of collaboration links between top active authors in each window respectively. Note, for these two networks only the top active authors in the respective time windows are included. The top active authors without any coauthors (hence singleton nodes) are removed, and the top active coauthor pairs with no other neighbors (in which case the denominator for $O_{ij}$ is 0) are also removed. Figure 9 shows the relationship between the link overlap and tie strength of the two coauthor networks respectively. We see that the coauthor network of top active authors presents a pattern in between the patterns in ordinary social network and general coauthor network respectively. On one hand, a similar trend as the ordinary social network is displayed (right hand side part of Figs. 9A and 9B). For links with large weight, the link overlap is also relatively large. This shows the general trend when doing research that many authors will collaborate when they study similar topics. On the other hand, there are author pairs in the same community but seldom collaborating (left hand side of Figs. 9A and 9B). Those authors might be research leaders working on the same specific topic. Each of them has many coauthors in the same area, but they do not collaborate often directly, since they need to compete with each other in order to get resources and support for their own research. Therefore, although they might have large overlap of coauthors, they do not collaborate

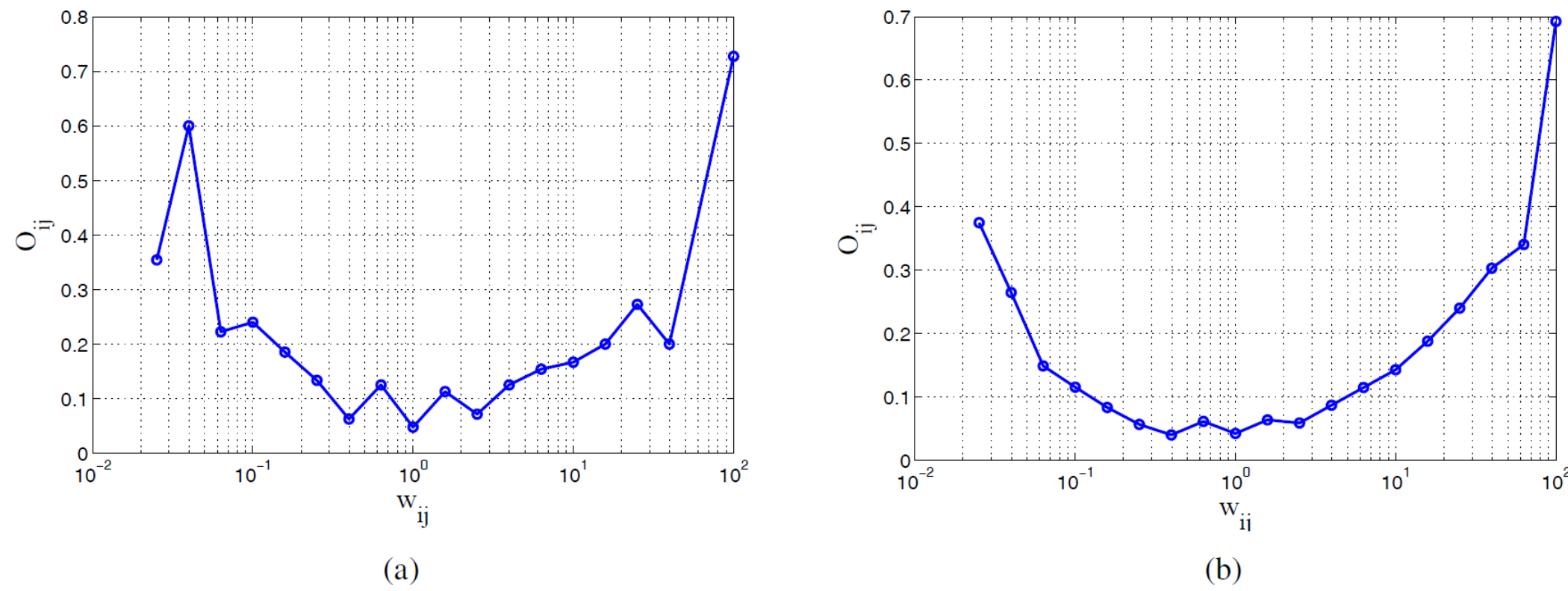

**Figure 9** **Relationship between link overlap and link weight in the two active windows.** We use logarithmic binning for $w_{ij}$ and choose the median value of $O_{ij}$ in a bin. (A) is the result in [1988, 1995] window. (B) is the result in [1998, 2009] window.

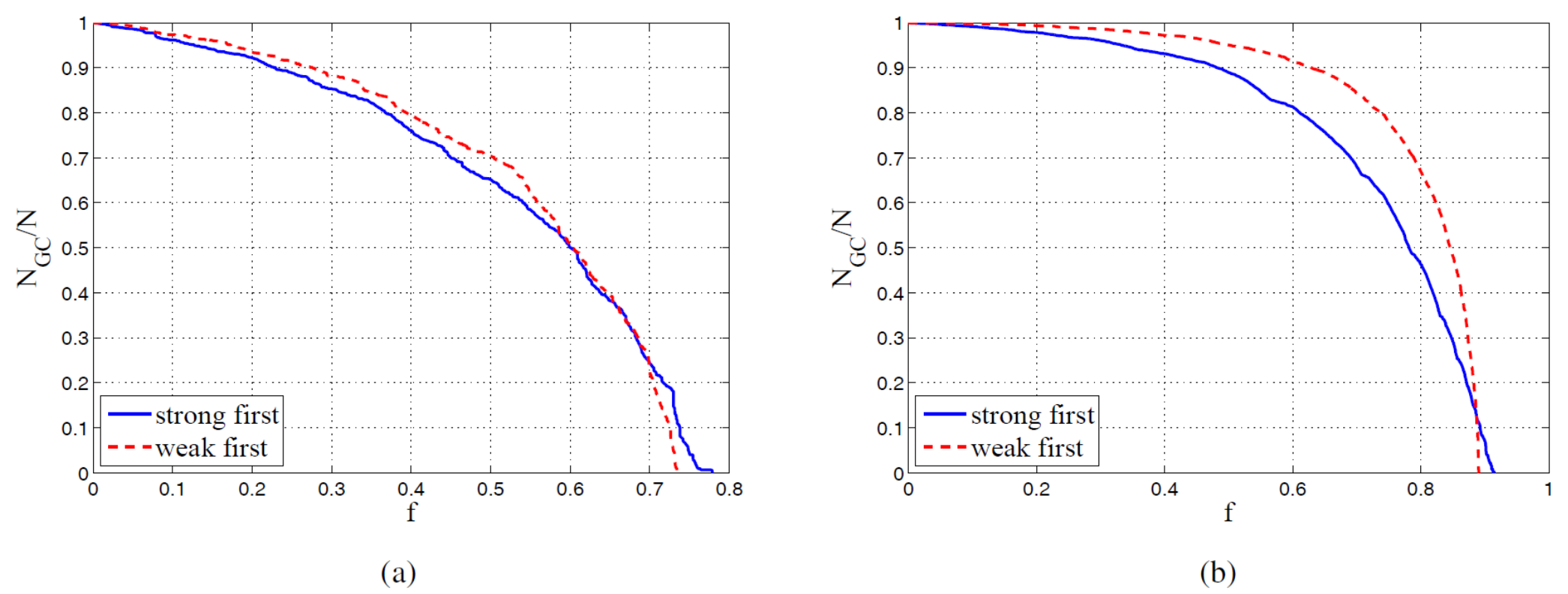

**Figure 10** **The relative size of the remaining giant component as a function of the fraction of removed links in the two active windows.** (A) is the result in [1988, 1995] window. (B) is the result in [1998, 2009] window.

much directly. There are also author pairs with very small link overlap but relatively large link weight (the middle part of Figs. 9A and 9B). This indicates the collaboration between different communities, thus the interdisciplinary research trend in Computer Science.

To confirm our conclusions above, we also do analysis on the giant component of each coauthor network of top active authors. We remove links one by one from the giant component based on the link weight, either in decreasing order or increasing order, and keep track of the size of the remaining giant component as a function of the fraction of removed links. The result is shown in Fig. 10. We see that in general, the giant component shrinks faster when the stronger links are removed first, which indicates that links are stronger between different communities. This phenomenon is similar to the case in general coauthor network, but they have quite different implications. For the general coauthor network, the weaker connection inside a community is due to the junior students in a research group (*Pan & Saramäki, 2012*), while for the coauthor network of top active authors, since each node represents a senior researcher, the relatively weaker connection

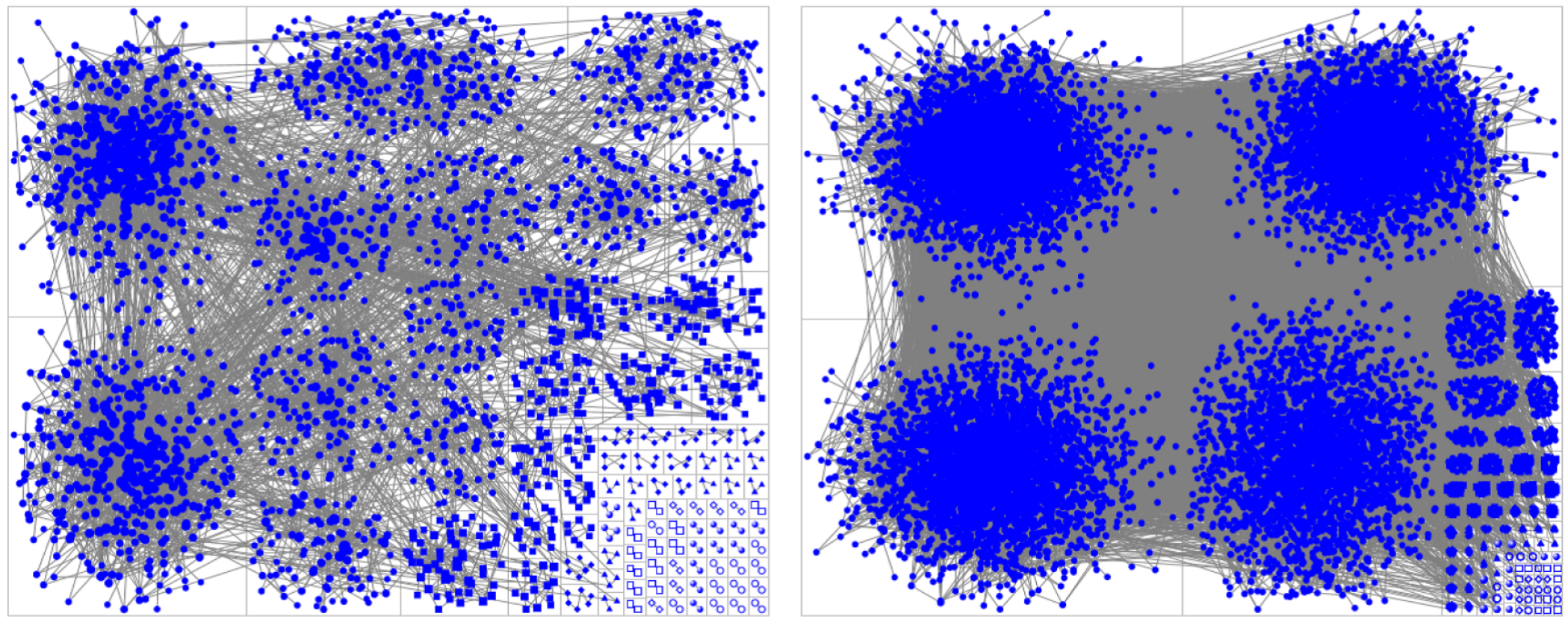

**Figure 11 Comparison of clustering results in the two active windows.** (A) is the result in [1988, 1995] window. (B) is the result in [1998, 2009] window.

inside a community is due to their competition in the same topic, and the stronger connection between different communities is due to the collaboration in interdisciplinary topics. Besides, by comparing Fig. 10A with Fig. 10B, we find that the gap between the two shrinking curves are becoming larger, and the giant component is also becoming more resistent to the removal of links. In Fig. 10B, most of the nodes still exist in the giant component even when half of the links are removed. The larger gap implies more strong connections and less weak connections between communities, which again shows the interdisciplinary research trend. The better resistance to link removal means that inside the same community, there are more links with the strongest/weakest connections. It indicates on one hand the collaboration of senior authors in the same topics, on the other hand the fierce competition among research leaders on similar topics.

## Topic trend based on top active author collaboration

Based on the leadership status of top active authors in their academic community, we can also get a sense of what have been the hot topics in the community, by observing the evolution of the research topics the top active authors work on.

For this study, we still take the two sets of top active authors in the active windows [1988, 1995] and [1998, 2009] respectively, and compare the differences between them. As with the previous analysis, we first build the two coauthor networks based on the existence of collaboration links between top active authors in each window respectively, and remove the top active authors without any coauthors, as they will not be part of any clusters anyway. We then apply the Clauset–Newman–Moore algorithm (*Clauset, Newman & Moore, 2004*) to do clustering for the two coauthor networks. This algorithm is based on the optimization of the network modularity, which measures when the clustering is a good one, in the sense that there are many edges within clusters and only a few between them. Figure 11 shows the clustering result for the two windows. Different clusters are put in different grids, with blue nodes representing top active authors in clusters. The grey lines between nodes in different grids represent the collaboration relationships

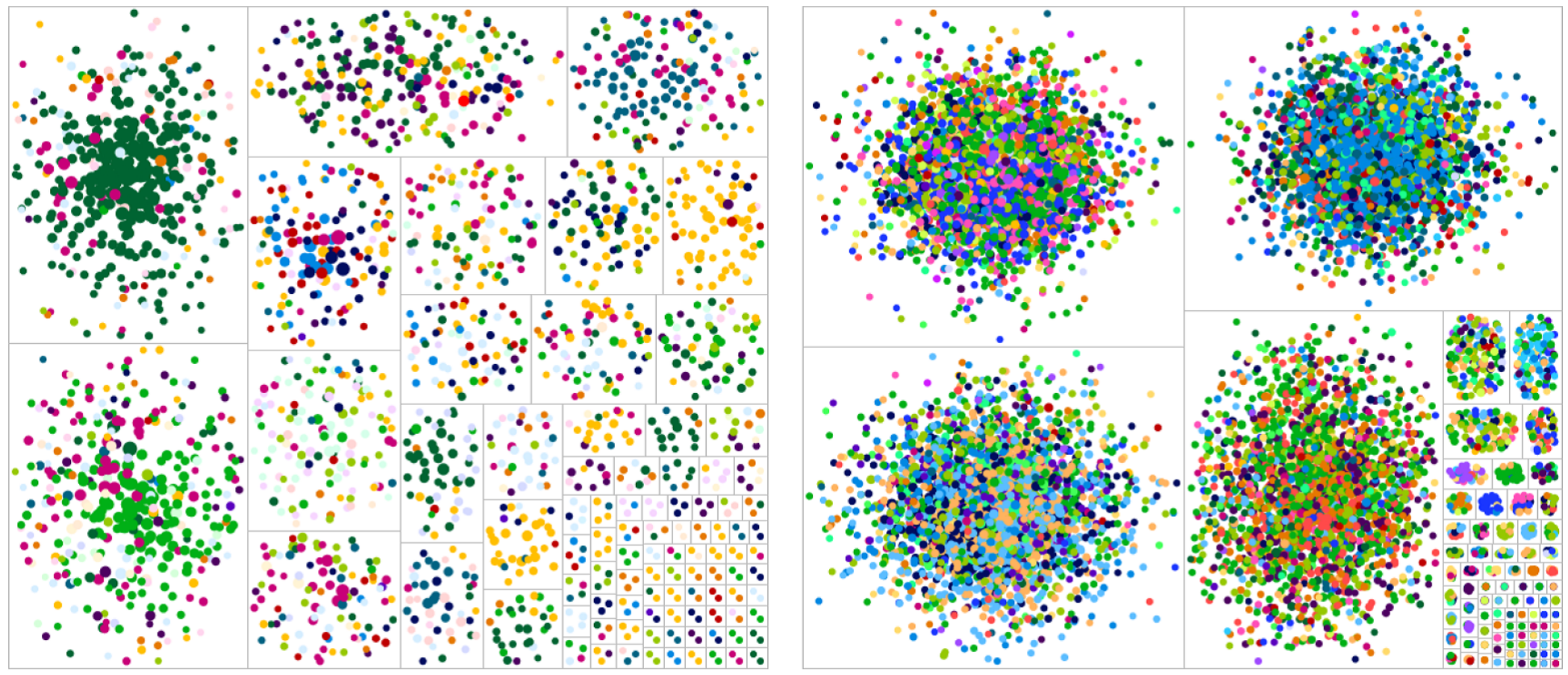

**Figure 12 Major subdomain of top active authors in the two active windows.** (A) is the result in [1988, 1995] window. (B) is the result in [1998, 2009] window.

among different clusters. We can see that in the earlier window, the clusters are more fragmented. The smallest cluster contains only two authors (the minimum possible size). The connections (represented by the number of links) between different clusters are not strong. Our interpretation is that in the earlier years, researchers tended to work more in isolation or with small scale (e.g., thesis mentor-mentee type of) collaboration, with little cross teams collaboration. The period [1988, 1995] is also before the advent of WWW, which might be attributed as an important factor of increased research collaboration. In the second window [1998, 2009], however, four largest clusters with similar sizes emerged and seemed to dominate all the other clusters in size. Moreover, many more collaboration links exist between different clusters. This indicates that Computer Science as a research field had become more interdisciplinary (at least within its field) with much more extensive collaboration among its researchers, which has been shown in previous analysis.

Since our clustering is conducted based on the existence of collaboration links, and through the publications of each top active author we extract their major research subdomains, we can visualize research as the mixing (or collaboration) of ideas from different research subdomains. For the years in the first window of time, a few research subdomains are the focus of research then, and many other research ideas were emerging and small research subdomains were just being formed. This is manifested by the large number of research clusters, the minimal collaboration between these clusters, and each cluster hosting a relatively homogeneous group of researchers. This is illustrated in Fig. 12A, where each node still represents a top active author, with a color representing the major research subdomain of that author (If an author publishes in multiple subdomains, then the major subdomain is the one most of that author's publications during the top active period belong to.). By the second window of years, a large number of top active authors belong to the four major clusters with heavy intra-cluster collaboration, with a relatively small fraction of top active authors still working in smaller clusters. Furthermore,

the four clusters are no longer so homogeneous, with a more mixed set of colors, as shown in Fig. 12B. Again, each node corresponds to a top active author, with a color representing the major subdomain that top active author works in, during the respective time window.

By now, you must be curious about what the large clusters in each of these two windows are. We show the answers in Figs. 13A and 13B for these two time windows. For each cluster, we show its composition in terms of the distribution of its researchers from the 24 subdomains of our metadata.[3] For the earlier time window [1988, 1995], the top three clusters are made up of mostly (1) "Algorithms and Theory" people, (2) "Databases" people, and (3) "Programming Language" people respectively. By the second time window, the top four dominating clusters are each hosting a more mixed set of top active authors, with the dominating subdomains being, respectively

(1) "Algorithms and Theory" and a set of application or technology areas, including "Networks and Communications", "Security and Privacy", "Computer vision", "Graphics" etc.
(2) "Databases" and "Artificial Intelligence" and some application or technology areas, including "Networks and Communications", "Human Computer Interactions", "Data Mining" etc.
(3) "Hardware and Architecture", "Software Engineering" and "Distributed and Parallel Computing", which may all be considered to be related to computing systems.
(4) "Artificial Intelligence", "Machine Learning and Pattern Recognition", "Multimedia", "Natural Language and Speech", and "Networks and Communications", which may all be considered to belong to multimedia technology, applications and systems.

These large clusters seem to map to the hot research areas and focus in Computer Science during those time periods.

From Fig. 13B, since there are more mixing of different subdomains in forming large clusters, we can also get a sense which subdomains tend to mix (collaborate) with others, and which subdomains tend to mix with each other. It seems "Networks and Communications", perhaps playing an infrastructure or glue role, tend to mix with others the most. "Artificial Intelligence" seems to mix mostly with "Machine Learning" and "Databases", which perhaps represent the "thinking" and "memory" aspects of artificial intelligence. Finally, it is also clear that the trend is for more and more interdisciplinary research, rather than for people in each subdomain working alone, which corroborates with previous findings.

## DISCUSSION

Our study is only based on publication and coauthorship data, so it is not possible to make meaningful assessment of research impact. A useful future direction is to incorporate the study of the research impact, based on whatever reliable measures for that, of top active authors and non-top active authors. This will help us further understand how good research results are achieved in the era of team research. Nonetheless, the insights from our study should still be helpful to policy makers in academia. For example, different

[3] We
Sci
to
riz

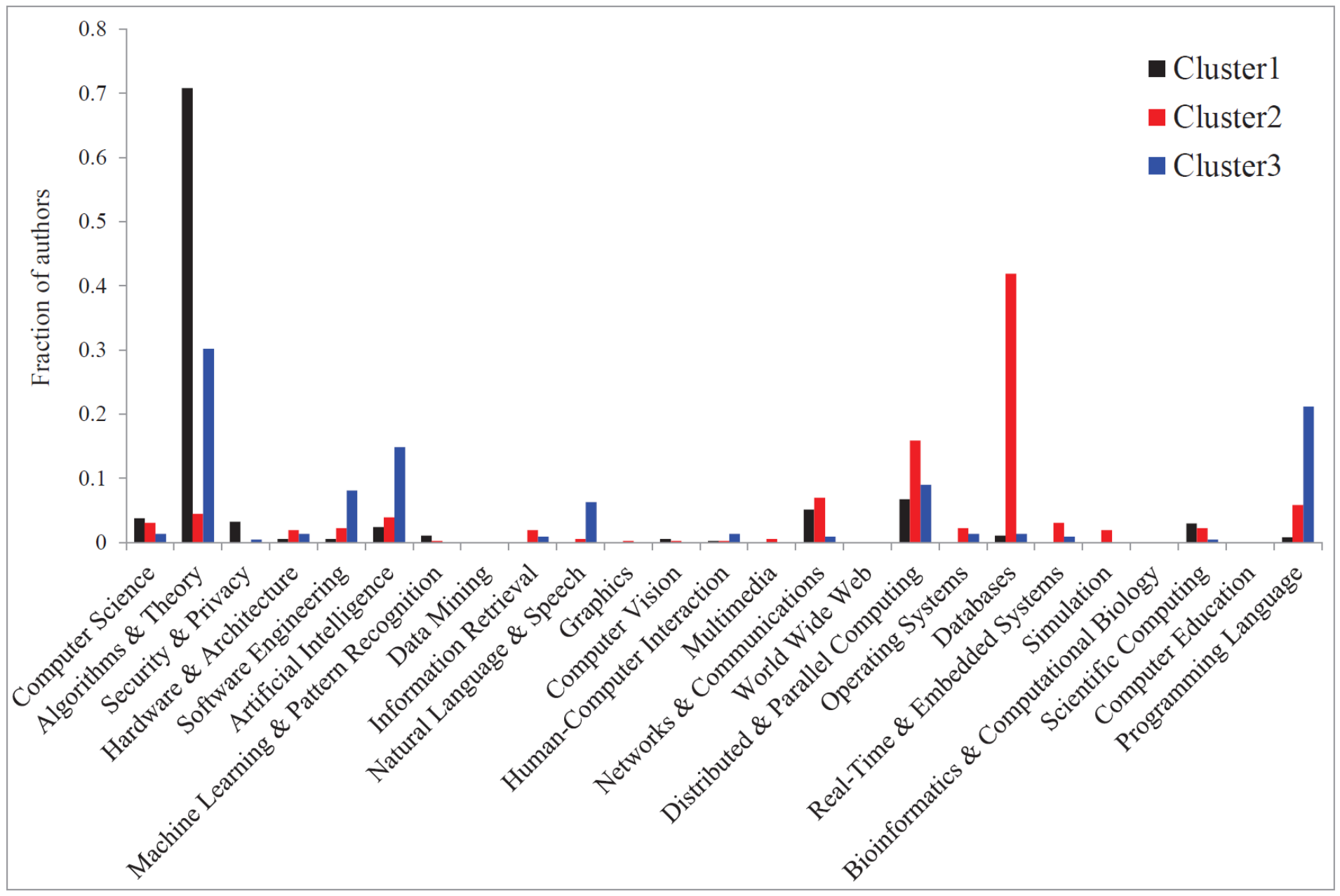

(a)

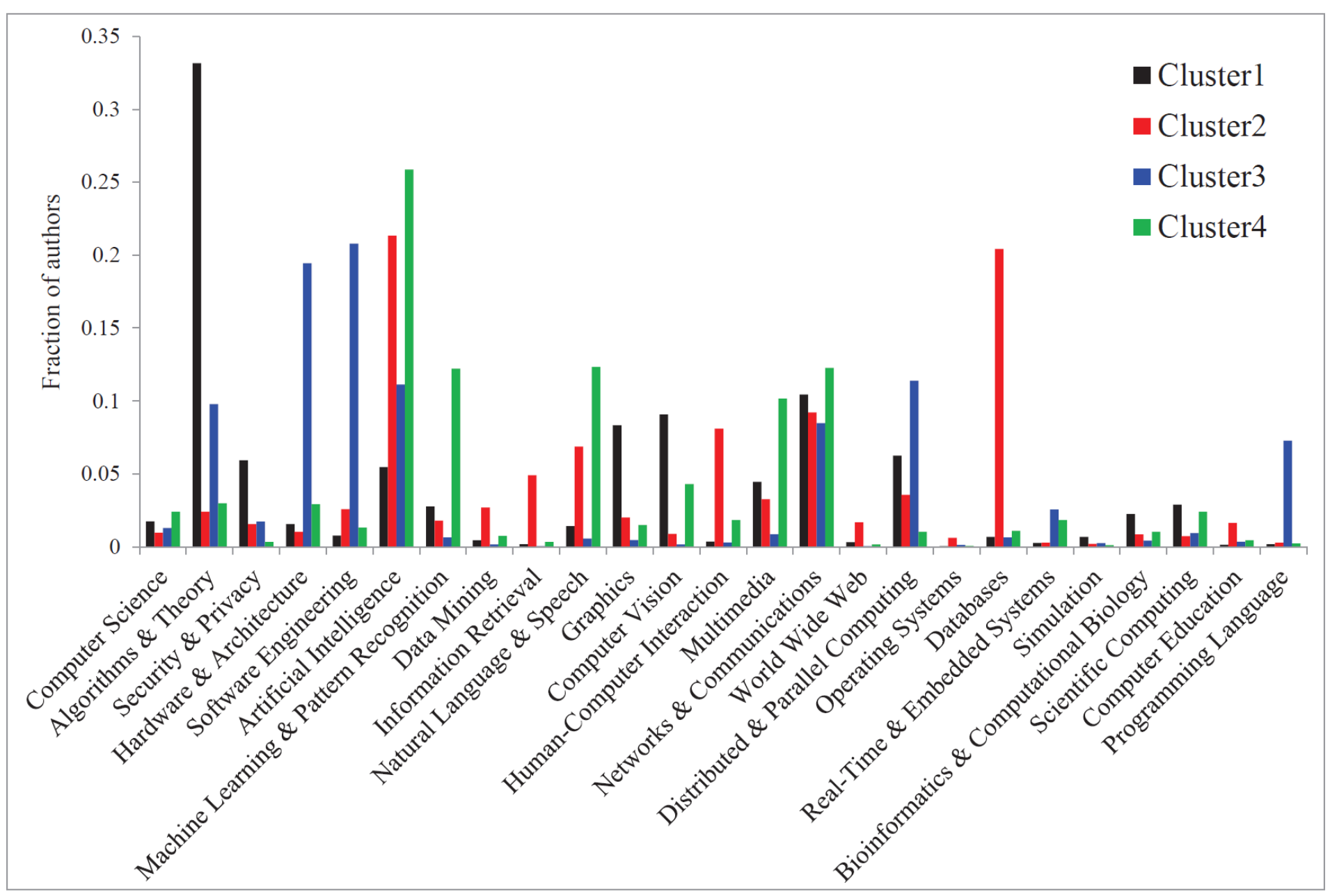

(b)

**Figure 13 Subdomain distribution in the two active windows.** (A) is the result in [1988, 1995] window. (B) is the result in [1998, 2009] window.

evaluation frameworks may be used for different types of authors to get proper assessment of the different roles in team research. By serving as the core of the community, and as leaders of research teams, the top active authors tend to easily build up his/her academic output quantitatively, compared to less active authors and newcomers. It is necessary to take into account of the different roles, to help better assess the individual contributions, and give proper recognition.

Besides, considering the evolution of different research topics in the community, we might have some mainstream topics which attract much attention from the community, just as the large clusters observed above. However, we may also have some topics which are relatively cold and studied only by a small group of authors. It is natural that funding institutions might tend to support research proposals on mainstream topics. However, the study on some cold topics should also not be ignored. They need to find a balance between the research topics studied by different scales of authors. And it is the governors' responsibility to make efficient resource allocations on different research topics based on their evolution patterns.

The evolution of research topics and the clustering of these topics may also have some interesting implications. On the one hand, it is encouraging to see more and more collaboration between authors leading to increasingly more interdisciplinary research, which, arguably leads to better research. On the other hand, it may also be the manifestation of the tendency by researcher to pursue hot topics, which is a less risky approach in a very competitive research environment. The policy makers may consider the proper balance between the convergence and diversity of researchers.

Nowadays, there are plenty of research funding encouraging research collaboration, not only between departments within a university, but across universities in a region, and also internationally, which is consistent with the trend of increased collaboration we observe. The nature of inter-team collaboration is worth further studies. On the one hand, it is possible that collaboration is mostly bringing together researchers of different backgrounds so they will complement each other. On the other hand, it is also possible that collaboration is bringing together teams working on the same topics (forming larger teams), which may reduce competition. The situation is likely the combination of both cases. An in-depth study of the trend and effectiveness of such collaboration behavior will also help funding policy makers.

Despite the insights we get from our findings, we understand that there also exist some limitations in our dataset. As mentioned by *Orduña-Malea et al. (2014)*, MAS has incomplete indexation of papers in recent years, which may result in a relatively smaller coverage of MAS than other popular bibliographic databases like ISI Web of Science and Scopus. But our focus in this paper is only the Computer Science field, not the general science research. And we only used data before 2010, the year when the incomplete indexation problem began to appear in MAS dataset (*Orduña-Malea et al., 2014*). With these restrictions, we think the data from MAS is good enough. An important reason for using the MAS data is that it comes with tagging of the papers by subdomains, which allows us to tag authors as well. This is a feature necessary for us to study the trend of

research topic clustering. The author name ambiguity problem is still a challenge to us. The MAS data has gone through some editing to remove some name ambiguity problems. Without involving authors to help correct ambiguity problems (as implemented by Google Scholar), the problem is not eliminated. However, in our study, since most analysis is at the statistical average level on different author sets, we believe the impact is not that severe. Besides, for the topic trend study, since only major subdomains are considered, this will also remove some bias caused by duplicate records of authors.

## CONCLUSION

In this paper, we took an analysis on a new set of authors, i.e., top active authors, who have uninterrupted and continuous presence in the scientific literature over a period of time. Thus top active authors may represent the most active researchers and serve as the core workforce in the community. We analyzed and compared the collaboration patterns and productivity of top active authors versus average authors in the Computer Science field. Results show that top active authors are serving as the core of the research community and the study of top active authors can help us have a better understanding of the general trend of team research in the community. We also studied the research topic trends by analyzing the evolution of the coauthor network structure of top active authors and the detailed research topics the top active authors work on. Results indicate that Computer Science, as a research field, is showing an increasing tendency for interdisciplinary research in the community. Our conclusions not only show the general trends in doing research in the academic community, but also provides insights for policy researchers and policy makers in academia to develop better evaluation methods and doing more efficient resource management in system.

Our analysis is just an initial attempt for the understanding and visualization of the general trend in the academic ecosystem. For future work, analysis on datasets in other research fields can be conducted and more measurements besides the ones we focused on in this paper can also be proposed. Theoretical analysis and modeling of the authors' team research patterns are of interest as well.

### Funding

The authors received no funding for this work.

### Competing Interests

The authors declare there are no competing interests.

### Author Contributions

- Yan Wu conceived and designed the experiments, performed the experiments, analyzed the data, wrote the paper, prepared figures and/or tables, performed the computation work, reviewed drafts of the paper.

- Srinivasan Venkatramanan and Dah Ming Chiu conceived and designed the experiments, wrote the paper, reviewed drafts of the paper.

## Data Availability

The data we used belongs to Microsoft Academic Search and is available upon request from Microsoft: http://datamarket.azure.com/dataset/mrc/microsoftacademic. Another project that made the data available (upon request) is at: http://cse.iitkgp.ac.in/resgrp/cnerg/. The latter source of data has been pre-processed for ease of use.

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
