# Peer review of "Research collaboration and topic trends in Computer Science based on top active authors"

_PeerJ Computer Science, doi:10.7717/peerj-cs.41_

## Round 0.1 · original submission · Major Revisions

Please observe that the reviewers have found that the amount of new material in this draft is not enough. A deep rewrite of the draft is needed so that the differences between the conference version and the journal paper are noticeable. There are also some important aspects of the experimental design that must be addressed.

·

Basic reporting

No Comments

Experimental design

Although it is said that for most experiments are presented only the results for a window due to space limitations, it would be a good idea showing, at least briefly, results of other windows, to see that the results are similar (as it is said in the paper).

Validity of the findings

No Comments

Reviewer 2 ·

Basic reporting

The authors present an interesting study on collaboration based on MAS data for the area of Computer Science. For this they compare two subset of authors: UCP authors vs. average authors. In general terms the paper is correct and written in a clear fashion. Although research collaboration is a widely studied topic, the data set employed allows to analyze a large dataset at the individual level, offering interesting insights on the collaboration and productivity patterns of researchers. However, there are several aspects which need of a more through description as well as the organization of the paper is a bit confusing.

Content

- The introduction includes details about the dataset (last paragraph) even before describing it. I would not describe such details at this point and mainly focus on the objectives and specific goals the paper aims at.

- Also, research collaboration is a highly studied topic in research policy, it would be desirable for the authors to relate their introduction to this topic in order to emphasize the importance of their findings later on. In this regard, the authors characterize UCP authors as senior researchers, probably leaders of their research teams, etc. Identifying different 'types' of researchers may allow to develop different evaluation frameworks, maybe the authors could ellaborate on this in their discussion.

- In the related work section again there are important contributions in the field of scientometrics which are missing such as the classical papers of Katz, Beaver or Crane on research collaboration.

- The dataset section should be much more detailed. A more comprehensive description should be given about MAS, its coveraged, the metadata offered, its evolution (it seems to have been unofficially terminated (http://dx.doi.org/10.1108/OIR-07-2014-0169)) as well as figures on the total number of authors identified for the time frame used, number of publications, etc. Wrapping up, a detailed description of the limitations of the data source and descriptive figures of the data set employed.

Structure

The structure of the paper is quite confusing and I believe it could easily adapt to the 'standard' sections recommended by peerj author instructions. I suggest dividing the dataset section into Methods (description of the data source and data retrieval and processing) and Data (descriptive figures on the dataset and description on the metadata). Then, the three analyses: 1) Comparison between UCP and average authors, 2) Collaboration patterns and 3) Topic analysis could all be subsection of the results section. Discussion would be a separate section and not a subsection as currently. I believe this would greatly improve the readibility of the paper.

Minor issues

As someone who didn't know about the UCP author concept I found the beginning of the paper a bit confusing. I suggest paragraph 2 could be inserted in paragraph 1 before the sentence 'The idea is that UCP authors are the core...'.

Experimental design

There are a serious of limitations on the data set that should be reported with regard to its reliability. I understand that the authors are working with a large dataset and hence, these should be minimized. In any case, they should be mentioned.

Other than that I have no further comments with regard to this section.

Validity of the findings

It would be desirable for the authors to make available their data, code, algorithms and results, to show transparency on their study and allow reproducibility.

Also the conclusions seem quite weak. I encourage the authors to reinforce their findings by relating to the wide corpus of literature existing in relation with research collaboration, profiles of researchers and their implications on research policy and evaluation

·

Basic reporting

The paper presents an study about the research activity in Computer Science. It analyzes the research collaboration and topic trends in that research field.

The work is not original, it was prior published in a conference (http://dl.acm.org/citation.cfm?id=2742015). The paper summited to PeerJ is identical to that one.

Experimental design

The dataset used was gathered from Microsoft Accademic Research, which is a not updated and incomplete data source. Authors also limited the analysis to 2009. Other bibliographical data sources as ISI web of Science or Scopus could have been used.

Validity of the findings

The paper presents some limitations:

- the first one is the dataset used.

- the second one, is the "uninterrupted and continuous presence" (UCP) concept. It considers authors publishing papers every year in the periods analyzed. It is an important restriction. UCP authors are not necessarily the most productive and more cited one. A study only focused in this set of authors is very biased. Authors argue incorrect address, saying things as:

"... By analyzing their activities, we can visualize the major trends of the whole community".

"... From the collaboration patterns of UCP authors, we can get more insight about the Team Research, in particular its correlation to resarch productivity".

"... UCP authors ara able to achieve more sustained research activity, much higher level of research output, and accelerated growth in research out-put, BECAUSE OF THEIR ABILITITY TO BUILD RESEARCH TEAMS" => The authors dare to say something very difficult to prove.

- third one, Why authors use two non consecutive periods ([1988-1995] and [1998-2009]) to analyze the collaboration among UCP authors? In this address, authors justify the rise in collaboration to the birth of the WWW. This conclusion is very light. The WWW enabled greater access to information and the possibility of more data, but this does not necessarily imply or justify an increase in collaboration. Other reasons have had to contribute to this increase.

Examples of UCP authors must be added for a better understanding.

Additional comments

The work is not original, it was prior published in a conference (http://dl.acm.org/citation.cfm?id=2742015). The paper summited to PeerJ is identical to that one.

For these reasons, I consider the paper must be rejected.

---

## Round 0.2 · accepted · Accept

The authors have addressed the concerns raised by the reviewers. The conference paper has been extended significantly and the new version includes a revised abstract, a new "discussion" section and some other changes and clarifications.

Reviewer 2 ·

Basic reporting

The authors have responded to all the comments and made the necessary changes.

Experimental design

No comments

Validity of the findings

No comments

Additional comments

No comments